# ResoNet: Noise-Trained Physics-Informed MRI Off-Resonance Correction

**Alfredo De Goyeneche**[1], **Shreya Ramachandran**[1], **Ke Wang**[1], **Ekin Karasan**[1],
**Joseph Cheng**[2], **Stella X. Yu**[1,3], **Michael Lustig**[1]
[1] Electrical Engineering and Computer Sciences, University of California, Berkeley
[2] Radiology, Stanford University
[3] Computer Science and Engineering, University of Michigan
{adg, shreyar, kewang, ekinkarasan, stellayu, lustig}@berkeley.edu
jycheng@stanford.edu, stellayu@umich.edu

## Abstract

Magnetic Resonance Imaging (MRI) is a powerful medical imaging modality that offers diagnostic information without harmful ionizing radiation. Unlike optical imaging, MRI sequentially samples the spatial Fourier domain ($k$-space) of the image. Measurements are collected in multiple shots, or *readouts*, and in each shot, data along a smooth trajectory is sampled. Conventional MRI data acquisition relies on sampling $k$-space row-by-row in short intervals, which is slow and inefficient. More efficient, non-Cartesian sampling trajectories (e.g., Spirals) use longer data readout intervals, but are more susceptible to magnetic field inhomogeneities, leading to off-resonance artifacts. Spiral trajectories cause off-resonance blurring in the image, and the mathematics of this blurring resembles that of optical blurring, where magnetic field variation corresponds to depth and readout duration to aperture size. Off-resonance blurring is a system issue with a physics-based, accurate forward model. We present a physics-informed deep learning framework for off-resonance correction in MRI, which is trained exclusively on synthetic, noise-like data with representative marginal statistics. Our approach allows for fat/water separation and is compatible with parallel imaging acceleration. Through end-to-end training using synthetic randomized data (*i.e.*, noise-like images, coil sensitivities, field maps), we train the network to reverse off-resonance effects across diverse anatomies and contrasts without retraining. We demonstrate the effectiveness of our approach through results on phantom and *in-vivo* data. This work has the potential to facilitate the clinical adoption of non-Cartesian sampling trajectories, enabling efficient, rapid, and motion-robust MRI scans. Code is publicly available at: https://github.com/mikgroup/ResoNet

## 1   Introduction

Magnetic Resonance Imaging (MRI) is a powerful medical imaging modality that enables the acquisition of diagnostic information of soft tissue without harmful ionizing radiation. MRI visualizes hydrogen spins, mostly in water and lipids that are subjected to a very strong yet homogeneous magnetic field. When spins are excited, they emit a signal at a frequency proportional to the magnetic field present. Spatial encoding is possible by applying additional so-called gradient fields, which vary linearly in space and result in a Fourier relationship between the position of the spins and the frequency spectrum of the acquired signal. Since position is mapped to frequency, any additional magnetic field inhomogeneities will result in incorrect mapping and lead to image distortions and undesired artifacts. Magnetic field inhomogeneity can occur due to magnetic susceptibility differences between air and tissue, such as near the sinuses and ear canals in the head or near the lungs and

bowel in the body. Also, due to different molecular environments, spins in water and fat see different magnetic fields, which results in the so-called chemical shift phenomenon in the signal frequency. The severity of these *off-resonance* effects depends on different imaging parameters, with a general trade-off between scan time or signal-to-noise ratio and the level of artifacts.

**1.1  MRI Acquisition**  While there are numerous intricacies to MRI acquisition, the process can be succinctly understood as the collection of data in the spatial frequency domain, or *k*-space, of the imaged object magnetization $m(\vec{r})$. The relationship between the image and *k*-space data $M(\vec{k})$ is defined by the Fourier relationship,

$$M(\vec{k}) = \mathcal{F}\{m(\vec{r})\} = \int_{\text{vol}} m(\vec{r})e^{-j2\pi(\vec{k}\cdot\vec{r})}dV. \tag{1}$$

Here, $\vec{r}$ is the position vector in the object space, and $\vec{k}$ are the coordinates in *k*-space.

The MRI system employs a *k*-space traversal strategy where data is acquired sequentially over time along lines or smooth *trajectories* $\vec{k}(t)$. The time-dependent signal $y(t)$ corresponds to samples of $M(\vec{k}(t))$, described by the equation [1]:

$$y(t, \vec{k}(t)) = M(\vec{k}(t)) = \int_{\text{vol}} m(\vec{r})e^{-j2\pi(\vec{k}(t)\cdot\vec{r})}dV \tag{2}$$

Figure 1 illustrates different two-dimensional $\vec{k}(t)$ trajectories. The *k*-space measurements are collected in multiple shots, called *readouts*, with the scan time increasing linearly with the number of readouts. In most clinical scans, *k*-space is sampled in a row-by-row *Cartesian* manner, which consists of many short readouts. However, these brief readout durations are relatively inefficient in covering *k*-space, resulting in prolonged scan times and susceptibility to motion artifacts. Researchers have developed more efficient non-Cartesian sampling trajectories, such as *Spiral*, which utilize longer but fewer readouts, allowing traversal of *k*-space in a significantly shorter time. However, these longer readouts make the trajectory more susceptible to off-resonance effects.

**1.2  Off-Resonance in MRI**  Off-resonance arises mainly from two sources: 1) undesired spatial variations in the main magnetic field, which cause spins to resonate at a frequency offset $\Delta f(\vec{r})$

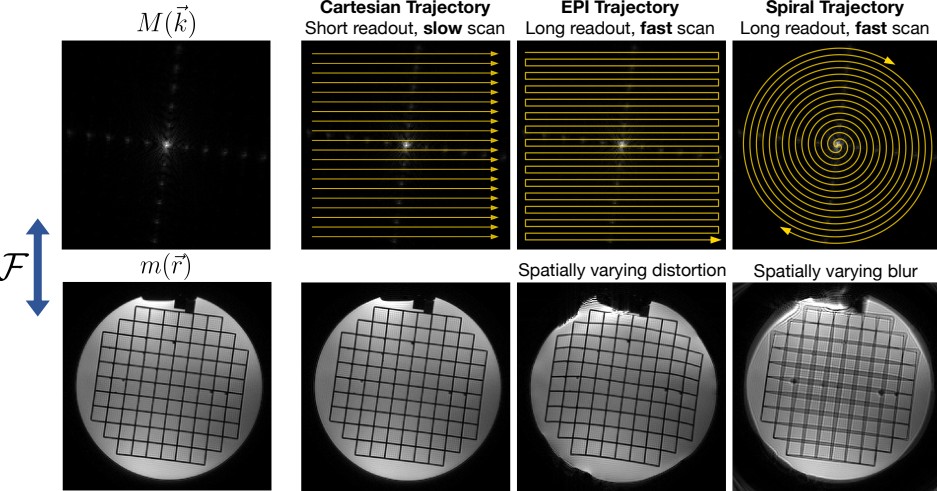

Figure 1: **Trade-offs between MRI acquisition strategies and off-resonance effects**: The MRI system directly samples the spatial Fourier domain (*k*-space) $M(\vec{k})$ of the object $m(\vec{r})$ in many shots, called *readouts*, each represented a by yellow arrowed path. The illustrated acquisition strategies assume a fully sampled *k*-space, where off-resonance effects depend on readout duration and geometry. Cartesian trajectories collect *k*-space samples in many short readouts in a row-by-row manner, with minimal off-resonance artifacts. Echo-planar imaging (EPI) and Spiral trajectories collect *k*-space samples via longer but fewer readouts in a significantly shorter total time. However, these are more susceptible to off-resonance effects, manifesting as image distortions in EPI and image blur in Spiral.

proportional to the field inhomogeneities, and 2) the inherent chemical properties of tissue types in the object, such as fat and water, where spins in fat tissue resonate at a fixed frequency offset $\Delta f_{\text{fat}}$ from those in water. These offsets are relative to a carrier frequency $f_0$ - the resonant frequency of hydrogen protons in water - which we omit in our equations for simplicity, presenting instead the baseband signal. Equation 2 assumed a homogeneous magnetic field and single tissue type, specifically water; however, since images contain both water and fat components and both components experience the spatially varying off-resonance $\Delta f(\vec{r})$, Equation 2 can be extended to

$$y(t, \vec{k}(t)) = \int_{\text{vol}} \left( m_{\text{water}}(\vec{r}) + m_{\text{fat}}(\vec{r}) e^{-j2\pi\Delta f_{\text{fat}}t} \right) e^{-j2\pi\Delta f(\vec{r})t} e^{-j2\pi\vec{k}(t)\cdot\vec{r}} dV. \tag{3}$$

In this equation, the additional exponential terms introduce space-varying linear phase modulation to image samples, where the range of the phase modulation increases with the readout duration, $t$, thus breaking the Fourier model. Therefore, reconstruction using an inverse Fourier transform will result in image artifacts that depend on the trajectory used.

In Cartesian trajectories, all the readouts are in the same direction, accruing linear phase across $k$-space. This results in image domain shift according to the Fourier shift property and thus to geometric distortion for space-varying inhomogeneities. Due to the short duration of Cartesian readouts, these off-resonance artifacts are negligible compared to those from trajectories with longer readouts, like Spirals (Figure 1). Please refer to [2] for a detailed derivation of the exact shift. Since Spiral trajectories go outwards from the center of $k$-space, the phase accrual is center-out. This results in spatial frequency components being displaced in different directions, leading to blurring and loss of sharpness in the image. The Point Spread Function (PSF) of this effect corresponds to a spatially varying blurring kernel proportional to the amount of local off-resonance. Please refer to the supplementary material for visual representations of these PSFs.

The mathematics underlying off-resonance blurring resembles that of optical blurring due to depth, as illustrated in Figure 2. The spatially varying magnetic field is mathematically similar to a spatial variation in depth, while the trajectory readout length is analogous to the size of the optical aperture. Just like larger apertures increase both SNR and blurring from depth variations, longer readouts increase both SNR and blurring from variations in resonant frequency. However, unlike blurring in optical systems, off-resonance blurring is additive, layering out-of-focus regions on top of those in focus. One could bring off-resonant regions into focus by demodulating the signal equation with a specific phase offset (Figure 2). For instance, extracting the fat phase term $e^{-j2\pi\Delta f_{\text{fat}}t}$ from the integral in Equation 3 and demodulating by it makes the fat image components sharp, leaving water components blurred. Also, MRI introduces an additional complexity known as partial volume effects [3], where a single voxel can contain both resonant (water) and off-resonant (fat) spins.

**1.3    Our Approach**    Off-resonance blurring is a system problem and has a known forward model, but solving for a direct solution is challenging due to its ill-posed nature. Recently, Deep Learning (DL) has enabled significant improvements over classical methods for solving inverse problems [5–8]. However, there are concerns over the black-box nature of end-to-end networks, overfitting, potential hallucinations [9], and the necessity of large representative training datasets. Physics-inspired DL methods, such as model-based unrolled networks [6, 10], mitigate these concerns by enforcing consistency to the acquired data with the known forward model of the system. Utilizing a forward model also enables the simulation of a training dataset, circumventing the need to collect large amounts of training data on an MRI scanner representative of all anatomies and contrasts.

In this work, we propose a physics-inspired unrolled-DL framework for off-resonance correction in MRI. The key contributions of this work are as follows:

- End-to-end training using only synthetic data: A distinguishing feature of our approach is the exclusive utilization of synthetic data with representative marginal statistics for network training (inspired by [11]). By leveraging synthetic random field maps, partial volume effects, coil sensitivities, and noise-like images, we eliminate the requirement for real MRI data during the training process. This approach allows us to overcome the limitations of acquiring large training datasets on an MRI scanner, which can be costly, time-consuming, and unfeasible in diverse scenarios.
- Estimation of a resonance frequency spectrum per voxel using a physics-informed DL framework: This effectively handles partial volume effects, i.e., separates fat and water contributions within each voxel. While we demonstrate results for a Spiral trajectory, we propose a generic framework that could be applied to other non-Cartesian trajectories.

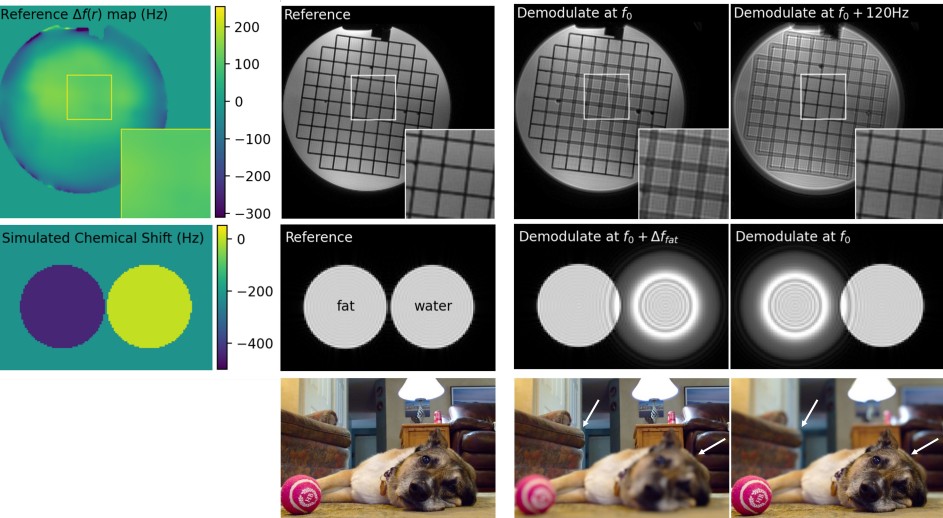

Figure 2: **Illustration of off-resonance blurring effects in Spiral MRI and analogy with Depth of Focus.** Top row: Water-based phantom data exhibiting off-resonance effects. The reference short-readout image required 4x the scan time. Middle row: a simulated example with non-overlapping water and fat circles. Demodulating at the water resonance frequency ($f_0$) brings the water circle into focus but causes blurring of fat, and vice versa. Bottom row: [4] Analogy with depth of focus in optical imaging, where focal point is sharp, while other depths are blurred. In MRI, different demodulation frequencies correspond to different focal lengths. Unlike optical blurring, off-resonance blurring is additive, with out-of-focus regions blurring on top of in-focus areas, making it challenging to create a seamless all-in-focus image by stitching together multiple demodulated images.

- Generalizability to multiple anatomies and contrasts: The proposed framework demonstrates strong generalizability, effectively correcting off-resonance artifacts across different anatomies and contrasts without the need for retraining.

**1.4   Related Work**   Recent developments in MRI highlight the potential of physics-driven DL methods in reconstruction tasks, spanning from physics-informed loss functions to unrolled network architectures [6, 10]. In off-resonance correction, traditional approaches include direct deblurring methods like autofocusing [12], which iteratively determines the sharpest image resonance frequency per voxel; however, these often rely on simplified sharpness criteria from optics [13], limiting their ability to generalize to more complex off-resonance scenarios encountered in MRI, such as partial volume effects. Model-based analytical methods, such as conjugate phase reconstruction [14], require a magnetic field map measured from additional scans. The field map model does not represent partial volume effects, introduces additional scan time, and is susceptible to motion artifacts; hence, this method often cannot completely correct for off-resonance [15].

More recently, data-driven DL-based approaches have been proposed to use convolutional neural networks (CNNs) to directly deblur the image [16–18] or estimate the field map from the image [19]. However, these approaches often neglect the physics of off-resonance blurring, do not handle partial volume effects, are limited in handling multi-coil acceleration, and rely on specific training data tailored to certain anatomies and contrasts.

## 2   Methods

Our proposed approach addresses the limitations of previous off-resonance correction methods by developing a forward model that handles partial volume effects and is compatible with multi-coil acceleration. By slicing the object into multiple frequency bins, we represent the tissue resonant at each frequency, leading to a comprehensive model that captures the full spectrum of frequency offsets. Our approach recognizes that off-resonance is a system problem intrinsic to the MRI acquisition process rather than a direct image feature problem. As a result, we leverage training on synthetic

random data, providing an effective strategy to address this challenge and improve generalization across different anatomies and contrasts.

**2.1 Physics-Informed Forward Model** The key in our approach is that we do not model the spatially varying field and fat shift directly. Instead, we extend the dimensionality of our image to $x(\vec{r}, \Delta f)$, which includes a frequency dimension $\Delta f$ such that:

$$m(\vec{r}) = \int_{\Delta f} x(\vec{r}, \Delta f) \, d\Delta f \tag{4}$$

We approximate this representation as a sum of multiple frequency bins $x_i(\vec{r}) = x(\vec{r}, \Delta f_i)$, such that $m(\vec{r}) \approx \sum_i x_i(\vec{r})$, with each bin resonating at a uniform frequency $\Delta f_i$ due to field inhomogeneities and water/fat frequency offsets. Relating back to the optical imaging analogy, this is equivalent to representing a 3D scene instead of a 2D scene with a depth map. Using this modeling and the linearity of the signal equation 3, we can express the contribution to the signal from each frequency bin $\Delta f_i$ as

$$y_i(t, \vec{k}(t)) = \int_{\text{vol}} x_i(\vec{r}) e^{-j2\pi \Delta f_i t} e^{-j2\pi \vec{k}(t) \cdot \vec{r}} dV, \tag{5}$$

where $y(t, \vec{k}(t)) = \sum_i y_i(t, \vec{k}(t))$. Since the phase modulation in a bin applies uniformly to the image bin content, it can be taken outside of the integral, *i.e.*,

$$y_i(t, \vec{k}(t)) = e^{-j2\pi \Delta f_i t} \int_{\text{vol}} x_i(\vec{r}) e^{-j2\pi \vec{k}(t) \cdot \vec{r}} dV. \tag{6}$$

Discretizing the Fourier Transform, we obtain

$$y_i = M_{\Delta f_i} E x_i, \tag{7}$$

where E is an encoding matrix implementing a Non-Uniform Fourier Transform (NUFFT), and $M_{\Delta f_i}$ is a diagonal matrix with elements $e^{-j2\pi \Delta f_i t}$ in the diagonal. $M_{\Delta f_i}$ phase modulates the $k$-space of the bin by the sample time and frequency offset.

Putting it all together, we can construct a complete forward model

$$y = \sum_i y_i = Ax, \tag{8}$$

where $x$ is the multi-frequency bin image, and A is the complete encoding matrix which combines the Fourier and modulation operators. Figure 3 illustrates our forward model.

It is important to note that for the sake of brevity and simplicity, we ignore in our description the use of multiple receiver coils in MRI, which readers can refer to in the following references [20, 21]. However, by using a general linear encoding operator $E$ it is trivial to extend our approach from a single receiver in which $E$ represents a Fourier operator to a multiple receiver case where $E$ also includes multi-channel receiver sensitivities.

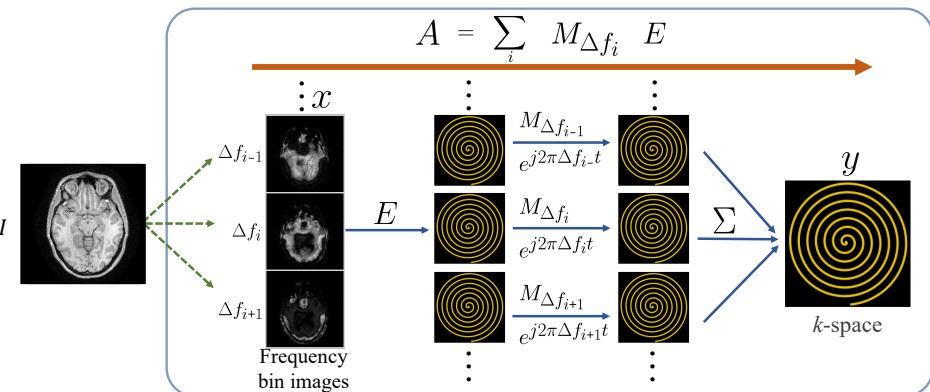

Figure 3: **Physics-informed forward model** $A$**:** The image is modeled as a stack of sharp images at different frequency bins. Each bin image is encoded using an encoding matrix $E$ (e.g., Non-Uniform Fast Fourier Transform (NUFFT)), and then phase modulated by $M_{\Delta f_i}$ corresponding to its bin frequency. Finally, the $k$-spaces across the bins are summed ($\Sigma$) to reconstruct the acquired $k$-space.

**2.2 Synthetic Training Data Generation** Inspired by [11], all the training data consists of simulated synthetic random field maps, fat/water partial-volume effects, coil sensitivities, and noise-like images. The statistics and smoothness of these synthetic images are designed to model those of real data. Specifically, we generate random images and field maps $\Delta f(\vec{r})$ by applying an FFT to exponentially weighted random complex data, where the weighting radius determines the level of smoothness. Additionally, to create image examples $I$, we combine multiple random images to obtain edges with various sharpness levels. Similarly, random sensitivity maps $S$ are generated from weighted random SPIRiT kernels [22].

To model the fat-water partial volume effects, we generate a random weight map $W$ that determines the percentage of fat vs. water at each voxel (Figure 4). From this weight map and image $I$, we obtain a fat image $I_{\text{fat}}$ and a water image $I_{\text{water}}$. Since fat has a resonant frequency offset from water, in addition to the random magnetic field map frequency offset $\Delta f(\vec{r})$, the fat image also includes a constant offset $\Delta f_{\text{fat}}$. Subsequently, the fat and water images are assigned to different frequency bins based on the random field map $\Delta f(\vec{r})$, fat offset $\Delta f_{\text{fat}}$, and the corresponding bin frequency values. This results in a multi-bin image representation $x$, fed into the forward model $A$ to obtain the simulated $k$-space data $y$.

**2.3 Model Architecture** To address the ill-posed nature of the problem, we employ a neural network for regularization. Inspired by the MoDL approach [6], we adopt an unrolled iterative model that is trained end-to-end. The input to the model is $A^H y$, a stack of images demodulated at multiple frequencies obtained by applying the adjoint operator $A^H$ on the raw $k$-space $y$. Regions of the object are sharp at the frequency bin corresponding to their local resonant frequency, while other off-resonant regions blur on top (Figure 5b). The objective of our model is to obtain a clean image at each frequency bin, as well as a combined output image, a water image, and a fat image, all without off-resonance blurring.

The unrolled model consists of Data Consistency (DC) modules and CNN-based proximal steps, as illustrated in Figure 5b. The objective function for the DC module is defined as follows:

$$x^k = \arg\min_x ||Ax - y||_2^2 + \lambda ||x - D_{\phi_k}(x^{k-1})||_2^2 \tag{9}$$

Here, $x^k$ represents the reconstructed image at iteration $k$, $A$ is the forward model, $y$ is the acquired data, $D_{\phi_k}$ denotes the CNN with learnable parameters $\phi_k$, and $\lambda$ controls the trade-off between data consistency and regularization. The objective aims to minimize the discrepancy between the forward model's prediction and the acquired data, as well as the difference between the reconstructed image and the output of the preceding CNN. From Equation 9, we can obtain:

$$(A^H A + \lambda I)x^k = A^H y + \lambda D_{\phi_k}(x^{k-1}), \tag{10}$$

which can be solved using the Conjugate Gradient (CG) method [23] to update $x^k$ at each iteration.

The CNN takes as input complex bin images with the real and imaginary components represented as separate channels. It includes residual blocks [24] and incorporates an attention module over

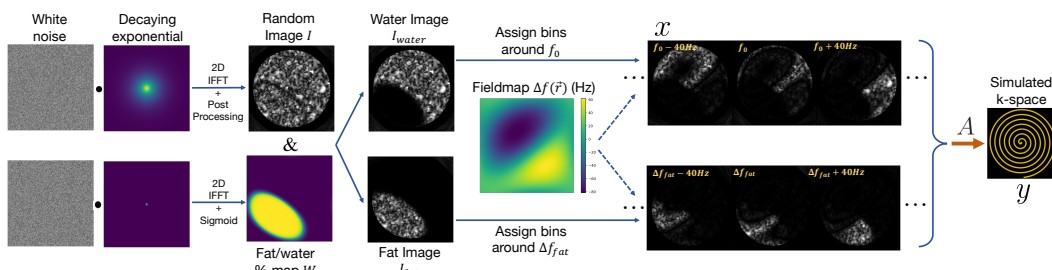

Figure 4: **Synthetic training data generation.** Random noise-like image $I$, fat/water percentage map $W$, off-resonance field map $\Delta f(\vec{r})$ are generated by applying an Inverse Fourier Transform to exponentially weighted white noise images, where the weighting radius controls smoothness. From $I$ and $W$, fat and water images, $I_{\text{fat}}$ and $I_{\text{water}}$, are obtained. Each voxel in these images is assigned to a frequency bin based on $\Delta f(\vec{r})$ and the fat frequency offset $\Delta f_{\text{fat}}$, obtaining our multi-frequency image $x$. Finally, $x$ is processed via the forward model $A$ to generate the $k$-space data $y$.

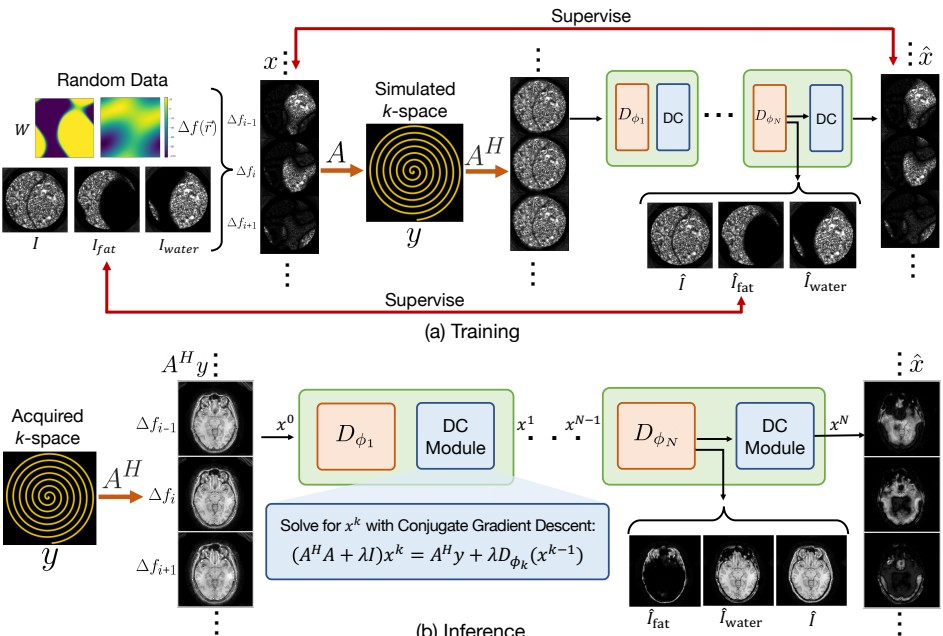

Figure 5: **Physics-informed deep learning framework for MRI off-resonance correction.** (a) Training process: The model is trained using randomly generated data, including images ($I$, $I_{\text{water}}$, $I_{\text{fat}}$) and frequency bins ($x$). Simulated k-space data $y$ is obtained by applying the forward model $A$ to $x$. The network, fed with images demodulated from $A^H y$, reconstructs frequency bins $\hat{x}$ and images $\hat{I}$, $\hat{I}_{\text{fat}}$, $\hat{I}_{\text{water}}$, and is directly supervised by the original data. (b) Inference process: Acquired k-space $y$ is processed through $A^H$ and fed to the model to reconstruct bins and images. The unrolled network comprises $N$ unrolls of CNNs and Data Consistency (DC) modules. Each DC module takes $A$, $y$, and previous CNN estimate $D_{\phi_k}(x^{k-1})$ to solve for $x^k$ using the Conjugate Gradient method.

frequency bins. This module allows the network to selectively focus on features from specific frequency ranges at each spatial location. Following the attention module, a series of residual blocks are employed. Finally, a convolutional layer produces the cleaner complex images at each frequency, which can then be fed into the DC module. Additionally, the network of the last unroll outputs a final combined image $\hat{I}$, a fat image $\hat{I}_{\text{fat}}$, and a water image $\hat{I}_{\text{water}}$.

**2.4 Model Training and Inference** During training, we utilize our synthetic data strategy to generate images $I$, $I_{\text{water}}$, $I_{\text{fat}}$, and the bin representation $x$, from which we obtain simulated $k$-space $y$ data using the forward model $A$. The unrolled model is fed with $A^H y$ as input, alongside $A$ and $y$ for the DC modules. We task the model with reconstructing the bin representation $\hat{x}$ and images $\hat{I}$, $\hat{I}_{\text{water}}$, and $\hat{I}_{\text{fat}}$, with direct supervision provided by the synthetic data (Figure 5a).

The forward model $A$ incorporates the acquisition characteristics of the readout trajectory in the NUFFT step, such as the type of non-Cartesian $k$-space trajectory, readout duration, number of interleaves, resolution, and field of view (FOV). The model learns to correct for a specific PSF associated with the trajectory, and a single model becomes versatile and adaptable to various anatomies and MRI contrasts, such as proton density (PD), $T_1$-weighted, and $T_2$-weighted images [1].

During inference, real acquired k-space data $y$, the forward model $A$, and $A^H y$ are provided to the network to reconstruct the bin representation and images (Figure 5b).

## 3 Experiments & Results

**3.1 Phantom and *In-vivo* Acquisitions** To evaluate the performance of our model, we acquired phantom data, as well as brain, knee, and abdominal *in-vivo* data using a GE 3T MR750W System. MR Pulse Sequences were designed using SpinBench Software, and data was acquired using RTHawk

v2.5.2 Software (Vista.ai Inc., Palo Alto, CA.). The *in-vivo* data was acquired from three volunteers who provided informed consent, and the study was conducted with IRB approval.

For the phantom, knee, and brain scans, we used a Spiral trajectory with 36 interleaves, a 5.2 ms readout duration, a resolution of $1\text{mm}^2$, and a 23 cm FOV. We acquired Proton Density (PD) weighted images for the knee and T1-weighted images for the brain. Alongside each scan, we acquired a reference scan with the same resolution and FOV but with a shorter readout (148 interleaves, 1.48 ms readout) to reduce artifacts. It is worth noting that for the knee scan, where we used a 2-second repetition time (TR), the acquisition time using the shorter readout trajectory is 5 minutes for a single slice, compared to 1.2 minutes for the longer readout sequence, resulting in a 4x speedup.

T1-weighted abdominal scans employed a 38-interleave Spiral trajectory with a 10.4 ms readout duration, a $1\text{mm}^2$ resolution, and a 35 cm FOV, with data acquired during breath holds. A short readout reference scan was also acquired (320 repetitions, 1.52ms readout duration); however, as breathing was permitted between scans, slices will not exactly align. With a 50ms TR, the long readout trajectory takes less than 2 seconds to acquire versus 16 seconds for the short readout one.

More details on Pulse Sequences, coils, and other parameters used are in the supplementary material.

**3.2   Model and Training Details**   In a 3T MRI, the fat frequency offset, $\Delta f_{\text{fat}}$, is reported to be approximately $-440$Hz [1, 25]. In our models, we simulated an off-resonance field map covering a range of $\Delta f(r) = \pm 180$Hz. To approximate the off-resonance spectrum, we utilized 21 frequency bins within the range of $-600$Hz to 200Hz, with a step size of 40Hz between bins. To better capture partial volume effects and avoid subtle data artifacts [26], data generation was performed at a high resolution, three times better than the reconstructed spatial resolution and ten times the reconstructed frequency bin resolution. Subsequently, the generated data was decimated in the Fourier domain to match the target resolution and dimensions.

The network architecture featured four unrolls. Each CNN module in these unrolls included three residual blocks, each containing two convolutional layers with 128 filters and a kernel size of 5, employing ReLU activation. The DC module used 12 conjugate gradient iterations with a trainable regularization parameter $\lambda$, empirically initialized to ensure that a ground truth input remained uncorrupted ($\lambda = 1$). We trained our model using PyTorch [27] in a Nvidia RTX3090 GPU, employing $\ell_1$ losses and the Adam optimizer [28]. We used the torchkbnufft library [29] for NUFFT operations. We pre-generated a training dataset to optimize memory usage and speed during training.

Two models were trained: one for the 23cm FOV trajectory, and another for the 35cm FOV scans.

In addition to our proposed approach, we aimed to replicate the methods in [16] and [17] to provide a relevant baseline for comparison, referred to as the "DL Baseline". We used the fastMRI brain dataset [30] with simulated Spiral trajectories and employed the network architecture and augmentation strategy proposed by [16] for training. It is important to note that this method only outputs a combined corrected image, without separate fat and water images or a spectrum across frequencies.

**3.3   Results on Phantom and *In-vivo***   The phantom and *in-vivo* results were obtained using models for specific trajectories and trained solely on noise data. Sensitivity maps for the acquired $k$-space $y$ were estimated using ESPIRiT [31] with the BART toolbox [32]. Though these maps may not always be perfectly accurate, they are fed to the model alongside the $k$-space, and our model remained robust to their potential inaccuracies.

Figure 6 showcases a phantom scan, a T1-weighted brain scan, a Proton Density (PD) weighted knee scan, and a T1-weighted abdominal scan, illustrating the capability of our approach to generalize across multiple anatomies and contrasts. The figures depict 1) the uncorrected long readout input data demodulated at the water frequency $f_0$, corresponding to the image a clinician would use for diagnosis, 2) the predicted water image output, 3) fat image output, and 4) combined clean output. Additionally, we compare our results against 5) the DL Baseline described in section 3.2 and 6) Autofocus [12] technique, which both aim to correct without the need for a separately acquired field map. Finally, we present 7) the uncorrected short readout reference image and 8) the reference field map obtained with separate scans. Please refer to the supplementary material for more detailed results showing all input and network output bins.

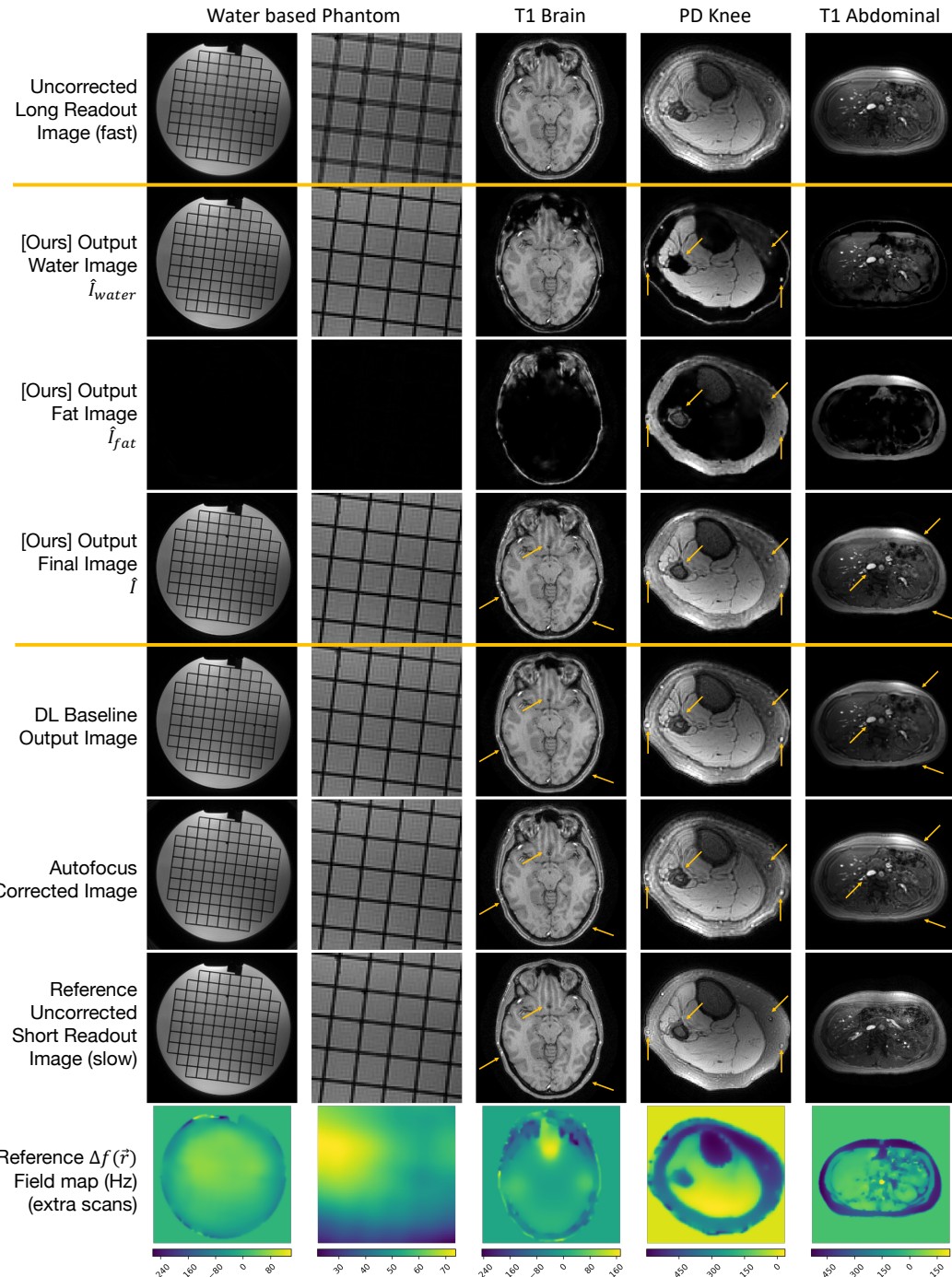

Figure 6: **Results of the proposed method on phantom and *in-vivo* data.** Results are presented for Water-based ACR phantom, as well as *in-vivo* T1-weighted brain, PD-weighted knee, and $T_1$-weighted abdominal scans. Top-to-bottom rows: 1) The uncorrected long readout input image, 2) output water image $\hat{I}_{\text{water}}$, 3) output fat image $\hat{I}_{\text{fat}}$, 4) combined output image $\hat{I}$. Reference baselines include 5) DL Baseline output image, 6) Autofocus [12] corrected image, 7) reference uncorrected short readout image (4x scan time for phantom, brain, and knee; 8x for abdominal scans). Lastly, 8) reference field map to visualize resonance frequency variations.

**3.4 Results on Simulated Off-Resonance Validation Set** While obtaining reference scans with shorter data readouts is possible, these scans will alter image contrast and introduce different artifacts. As a result, obtaining directly comparable ground truth data to compute meaningful quantitative metrics is infeasible. Therefore, as a performance proxy, we conducted an evaluation on a validation set created using the fastMRI brain dataset [30]. This set consists of brain anatomy images that were corrupted with simulated random off-resonance field variations and fat/water partial volume effects. The validation set comprised 1000 examples.

We compared the performance of our proposed approach and the DL Baseline by measuring the NRMSE (Normalized Root Mean Squared Error) and PSNR (Peak Signal to Noise Ratio) on the magnitude of the combined output images within the validation set. We report average metrics and their standard deviation over the 1000 examples in Table 1. Despite our approach being trained solely on synthetic non-anatomy data, it outperforms the DL Baseline regarding NRMSE and PSNR.

| Method \ Metric | NRMSE ($\downarrow$) | PSNR ($\uparrow$) |
|---|---|---|
| Our approach | **0.016 $\pm$ 0.011** | **37.1 $\pm$ 4.7** |
| Deep Learning (DL) Baseline | 0.049 $\pm$ 0.028 | 27.7 $\pm$ 5.7 |

Table 1: **Performance comparison on simulated off-resonance validation set.** NRMSE (lower is better) and PSNR (higher is better) for our method and the DL Baseline on a synthetic validation set of fastMRI [30] brain images with simulated random off-resonance and partial volume effects. Metrics show the average and standard deviation over 1000 examples.

## 4 Discussion

In this work, we propose a physics-inspired unrolled DL framework for off-resonance correction in MRI, showcasing its effectiveness through training solely on synthetic data. This approach addresses the challenges of collecting real MRI datasets for off-resonance correction, which can be unfeasible. Off-resonance blurring varies greatly across different anatomies due to differences in magnetic field variations, as seen in Figure 6. Our *in-vivo* evaluation showcases the generalization capabilities of our model across various contrasts and anatomies. Correcting off-resonance in MRI is particularly valuable for enabling non-Cartesian trajectories, such as Spirals, which significantly reduce scan time, leading to fewer motion-induced artifacts, shorter MRI exams, and improved clinical throughput.

The proposed framework and training strategy could bring further value to fields outside of MRI. The mathematics that arises from off-resonance correction in MRI is a generalization of lens aberration and depth from focus problems in microscopy and astronomy [33, 34]. The proposed technique could be adapted to these cases where there is a system error, like lens imperfection or atmospheric blurring.

**4.1 Limitations and Future Work** Despite promising results, our framework has limitations and areas for future improvement. Currently, networks are trained for specific trajectories, and future work involves handling multiple trajectories within a single model. This will include diverse trajectory patterns (such as Rosette [35], Looping Star [36]), resolutions, and FOVs. Another aspect for future improvement is simulating more extreme magnetic field map variations during training and increasing forward model resolution and network capacity, allowing for correction in settings such as regions close to air in the ear canals and nose. Currently, performance degrades when moving beyond the distribution of the training data simulation range.

## 5 Conclusion

In conclusion, we have presented a physics-informed unrolled DL framework for off-resonance correction in MRI. Our approach incorporates data consistency using a multi-frequency forward model that represents a resonance frequency spectrum for each voxel. We have bypassed the challenges of collecting large training datasets by training our network solely on synthetic data.

We have demonstrated results in generalization and performance across diverse anatomies and contrasts. By addressing off-resonance issues, our framework has implications for accelerating scanning, increasing clinical throughput, and making MRI more accessible. Continued research and refinement in this area will further contribute to the advancement of MRI technology and its application in healthcare.

## Acknowledgments

We acknowledge support from GE Healthcare and NIH grants R01EB009690 and U01EB029427. The authors would like to thank Suma Anand, Justine Taylor, and Ana Cismaru for their help with the paper editing.

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
