# ResoNet: Noise-Trained Physics-Informed MRI Off-Resonance Correction

# –

# Supplementary Material

**Alfredo De Goyeneche**[1], **Shreya Ramachandran**[1], **Ke Wang**[1], **Ekin Karasan**[1],
**Joseph Cheng**[2], **Stella X. Yu**[1,3], **Michael Lustig**[1]
[1] Electrical Engineering and Computer Sciences, University of California, Berkeley
[2] Radiology, Stanford University
[3] Computer Science and Engineering, University of Michigan
{adg, shreyar, kewang, ekinkarasan, stellayu, lustig}@berkeley.edu
jycheng@stanford.edu, stellayu@umich.edu

## A    Appendix: MRI off-resonance effects and Point Spread Function (PSF)

In magnetic resonance imaging (MRI), $k$-space data represents the spatial frequency content of the imaged object and is obtained through a process known as Fourier encoding. The choice of the $k$-space trajectory influences various aspects, such as acquisition time, image quality, and sensitivity to off-resonance effects.

For the following analysis, we will consider a water-only object and ignore partial volume effects due to fat tissue. The $k$-space sampling process described by the equation Equation (3) in the main paper becomes [1]:

$$y(\vec{k}(t), t) = \int_{\vec{r}} m(\vec{r}) e^{-j2\pi\Delta f(\vec{r})t} e^{-j2\pi(\vec{k}(t)\cdot\vec{r})} d\vec{r} \tag{11}$$

Here, $y(t, \vec{k}(t))$ represents the acquired $k$-space samples at time $t$ for a specific trajectory $\vec{k}(t)$. $m(\vec{r})$ denotes the object magnetization, and $\Delta f(\vec{r})$ accounts for the field inhomogeneity across space. The integral integrates over the object space, allowing the encoding of spatial information into the acquired $k$-space data.

The term $e^{-j2\pi\Delta f(\vec{r})t}$ in Equation 11 results in phase accrual along the readout of the trajectory. The amount of phase accrual depends on the field inhomogeneity $\Delta f(\vec{r})$. By analyzing the phase accrual map in the $k$-space domain, we can understand its effect in the image domain. Specifically, performing an inverse Fourier transform of the phase map allows us to obtain the Point Spread Function (PSF) associated with the trajectory. This PSF characterizes the off-resonance effects in the image domain.

When employing a Cartesian trajectory, the off-resonance effects primarily manifest as a shift in the reconstructed image. As all $k$-space samples are acquired along linear trajectories in the same direction, the accrual of phase in the frequency domain corresponds to a shift in the image domain according to the Fourier shift property. The shift amount depends on the space-varying inhomogeneities, with larger frequency offsets leading to greater shifts.

To visualize the off-resonance effects in a Cartesian setting, Figure 7 shows two constant field map $\Delta f(\vec{r})$ examples. In the top row, as the $k$-space data is acquired along the readout direction (horizontal axis), the phase increases linearly from left to right for each $k_y$ value. The middle row displays the respective PSFs, which shows the shift effect on the image domain. Finally, in the bottom row, we present the overall effect on a simulated brain image scan, which shows the brain shifted by different

37th Conference on Neural Information Processing Systems (NeurIPS 2023).

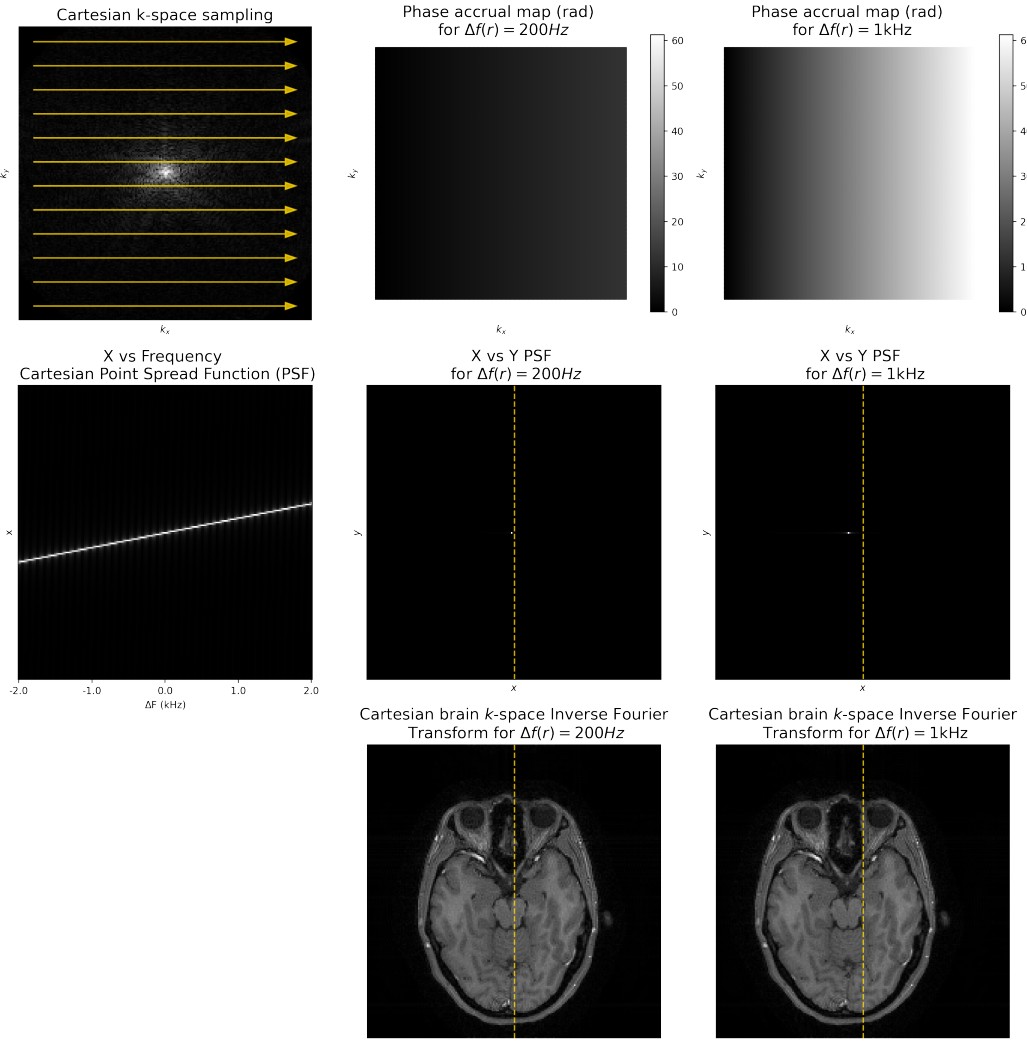

Figure 7: **Off-resonance shift in a Cartesian trajectory.** Left column: Cartesian $k$-space sampling trajectory and PSF in the $x$ vs. $\Delta f$ plane demonstrating the relationship between off-resonance frequency and image shift. Top row: phase accrual along the readout direction (horizontal axis) for two frequency offsets, with the phase increasing linearly from left to right for each $k_y$ value. Middle row: Point Spread Functions (PSF) obtained by inverse Fourier transforming the phase maps, revealing the shift effect on the image domain. Bottom row: example brain scan image with different amounts of shift corresponding to the selected frequency offsets.

amounts corresponding to the selected frequency offsets. Additionally, we present the PSF in the $x$ vs. $\Delta f$ plane, illustrating the relationship between off-resonance frequency and image shift.

On the other hand, when using a Spiral trajectory for $k$-space sampling, the off-resonance effects result in blurring in the reconstructed image. The amount of blurring depends on the off-resonance frequency, with larger frequencies leading to greater blurring effects.

To visualize the off-resonance effects in a Spiral setting, Figure 8 shows two frequency offsets. In the top row, as the $k$-space data is acquired along the Spiral readout, the phase increases linearly along the trajectory, resulting in larger phase accumulation as we move away from the center of $k$-space. The middle row displays the respective Point Spread Functions (PSF), showing the blurring kernels in the image domain. Finally, in the bottom row, we present the overall effect on a simulated brain image scan, which shows the brain with different levels of blurring corresponding to the selected frequency

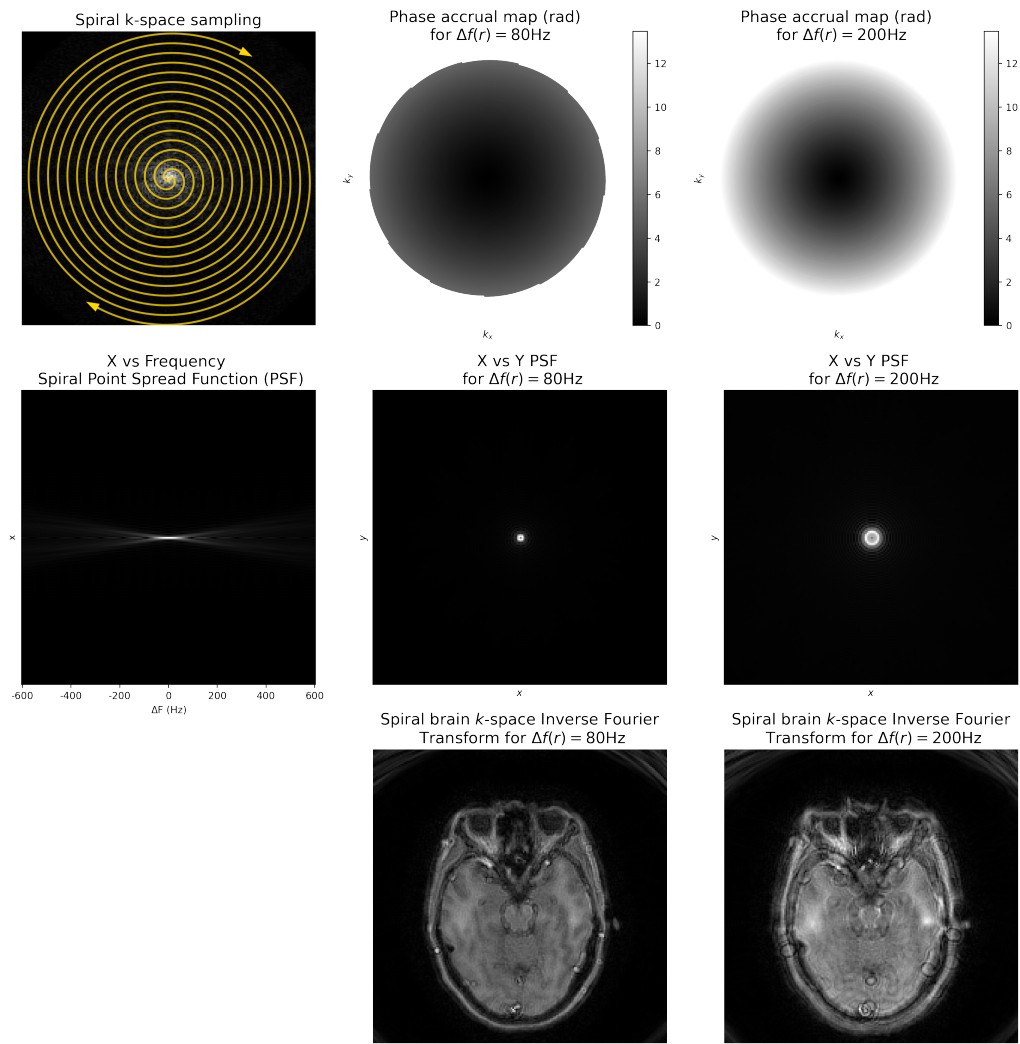

Figure 8: **Off-resonance blurring in a Spiral trajectory.** Left column: Spiral $k$-space sampling trajectory and PSF in the $x$ vs. $\Delta f$ plane demonstrating the relationship between off-resonance frequency and blurring kernel. Top row: phase accrual along the Spiral readout, with the phase increasing linearly along the trajectory and resulting in a center-out phase accrual in $k$-space. Middle row: Point Spread Functions (PSF) obtained by inverse Fourier transforming the phase maps, revealing the blurring kernels on the image domain. Bottom row: example brain scan image with different amounts of blurring corresponding to the selected frequency offsets.

offsets. Additionally, we present the PSF in the $x$ vs. $\Delta f$ plane, illustrating the relationship between off-resonance frequency and the size of the blurring kernel.

In the case of Spiral trajectory, we acquire $k$-space in fewer readouts and an overall faster scan by traversing $k$-space in longer readout durations. Since Cartesian off-resonance artifacts are less apparent, when illustrating the shift effect (Figure 7), we use larger field map offsets compared to the Spiral trajectory (Figure 8).

# B  Appendix: Expanded results

We provide a more detailed presentation of the qualitative results discussed in Section 3.3. Figures 9, 10, 11 and 12 showcase the qualitative results for the water-based phantom scan, T1-weighted brain scan, Proton Density (PD) weighted knee scan, and T1-weighted Abdominal scan respectively. These scans showcase different contrasts, anatomies, and fat/water constituencies.

The figures present the frequency bin demodulation inputs and frequency bin outputs, as well as the output water image, output fat image, and combined output image. We also include the reference uncorrected short readout image, reference acquired $\Delta f(\vec{r})$ field map, and the reconstruction obtained using the Autofocus [2] method. It can be seen that output bins correspond to the reference field map and that the model is able to separate fat and water components from the single Spiral readout.

For the phantom, knee, and brain scans, we used a Spiral trajectory with 36 interleaves, a 5.2 ms readout duration, a resolution of $1\text{mm}^2$, and a 23 cm FOV. For the T1-weighted brain scan, we used an Inversion Recovery sequence that included SLR excitation with a slice thickness of 2 mm. These scans were performed using a 22-channel head-neck coil (GE Healthcare, Waukesha, WI). For the knee scan, we obtained Proton Density (PD) images utilizing a Gradient Echo (GRE) sequence with SLR excitation and a slice thickness of 2 mm. In this case, we used a 16-channel Flexible knee Coil (NeoCoils, Pewaukee, WI). For the short readout images, we acquired data with the same resolution and FOV, using 148 interleaves with a 1.48 ms readout duration.

Abdominal scans used a separate Spiral trajectory of 38 interleaves, a 10.4 ms readout duration, a resolution of $1\text{mm}^2$, and a 35 cm FOV. Breath-held data was acquired with a GRE sequence with a slice thickness of 4mm and a 32-channel Anterior/Posterior Coil Array (GE Healthcare, Waukesha, WI). Similarly, a short readout reference scan was acquired (320 repetitions, 1.52ms readout duration). It is important to note that breathing was allowed between scans, so the reference scan will not be the exact same slice.

The echo times (TE), repetition times (TR), and inversion times (TI, if applicable) for each scan are specified on each figure.

## B.1  Autofocus reference

Both the Autofocus technique and our proposed method provide off-resonance correction without the need for a separately acquired field map. Autofocus achieves this by exhaustively determining the demodulation frequency that minimizes a specific metric at each voxel. This metric is the imaginary component of the image after removing the low-frequency phase across the object, which inversely relates to the sharpness of a voxel. We used demodulations within the range of $\pm 600$Hz, with a step size of 10Hz and a window size of 40 pixels.

## B.2  Reference $\Delta f(\vec{r})$ map

To obtain the reference magnetic field map, we performed two additional scans using a Gradient Echo (GRE) pulse sequence, which requires additional acquisition time. The GRE images were acquired sequentially at different echo times (TE), allowing us to obtain the phase evolution of the spins over the echo time difference. By subtracting the relative phase between these two images, we obtain the phase difference map denoted as $\Delta\phi$. We then scale this phase difference map by the echo time difference ($\Delta TE$) to obtain the field map:

$$\Delta f(\vec{r}) = \frac{\Delta\phi(\vec{r})}{2 \cdot \pi \cdot \Delta TE} \tag{12}$$

For our experiments, we used a $\Delta TE$ of 1ms and employed a Cartesian trajectory for $k$-space sampling. We also applied total variation denoising [3] to the displayed field map. It is important to note that the obtained field maps represent a single value per voxel and do not account for partial volume effects or provide a spectrum of values.

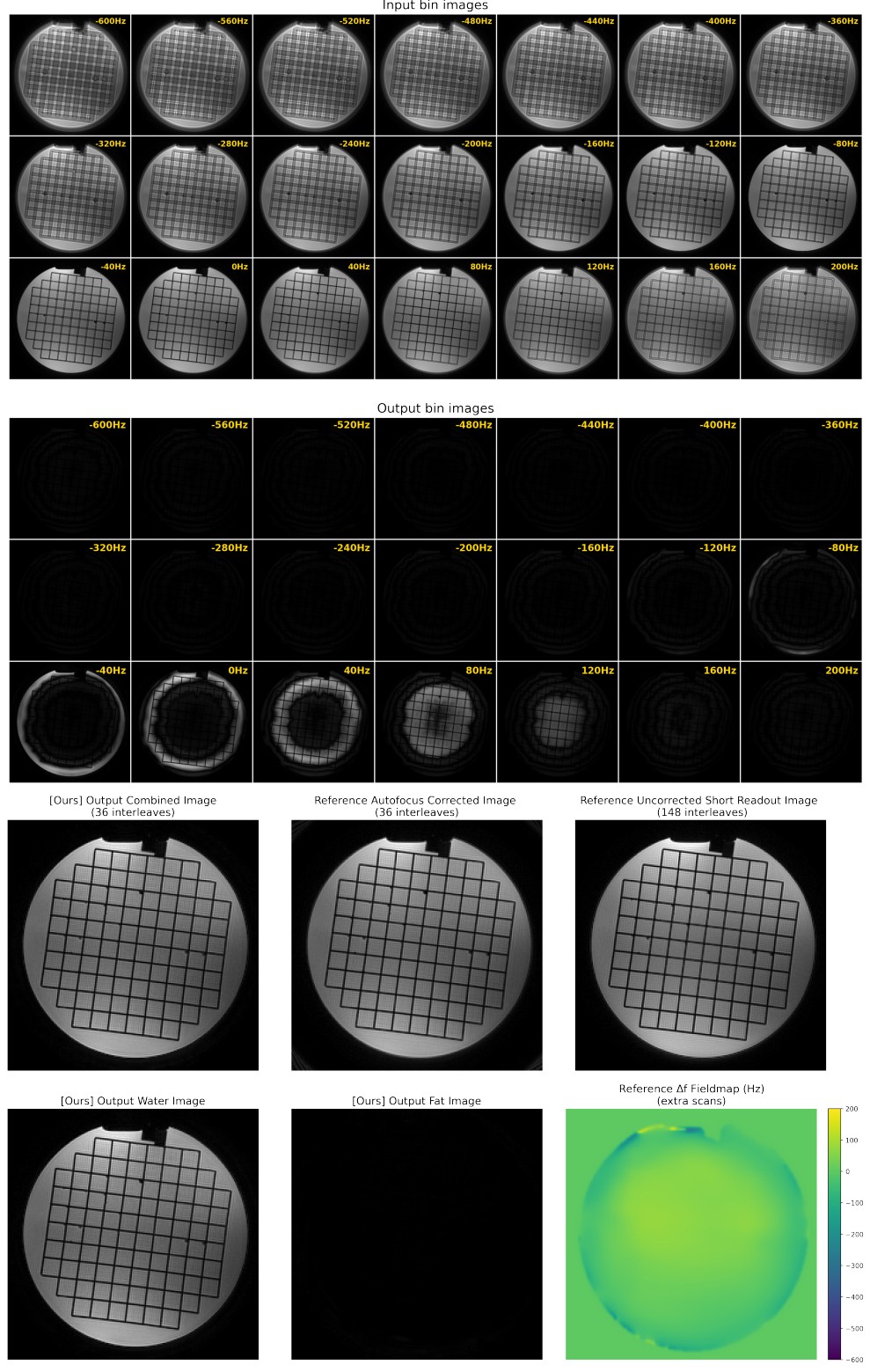

Figure 9: **Results of the proposed method on the water-based phantom scan.** Acquired using Gradient Echo (GRE) sequence (TE=4ms, TR=250ms, FA=60°). Top: Frequency bin demodulation inputs. Middle: Frequency bin output images. Bottom: 1) output combined fat-water images, 2) corrected image using Autofocus [2], 3) reference uncorrected short-readout image (4x scan time), 4) output water image, 5) output fat image, 6) reference off-resonance frequency map.

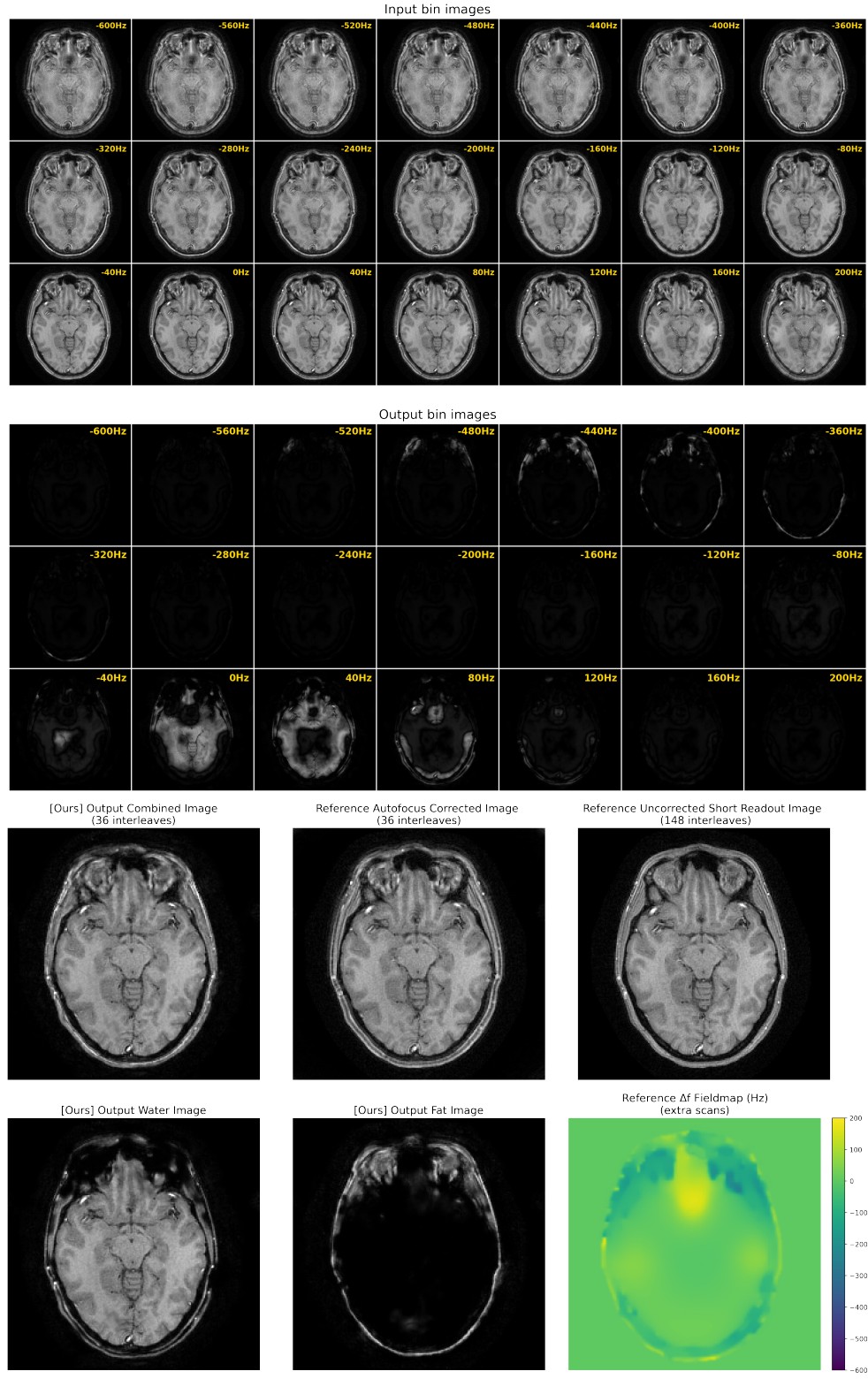

Figure 10: **Results of the proposed method on *in-vivo* T1-weighted brain scan.** Acquired using Inversion Recovery sequence (TI=1000ms, TE=4ms, TR=1500ms, FA=90°). Top: Frequency bin demodulation inputs. Middle: Frequency bin output images. Bottom: 1) output combined fat-water images, 2) corrected image using Autofocus [2], 3) reference uncorrected short-readout image (4x scan time), 4) output water image, 5) output fat image, 6) reference off-resonance frequency map.

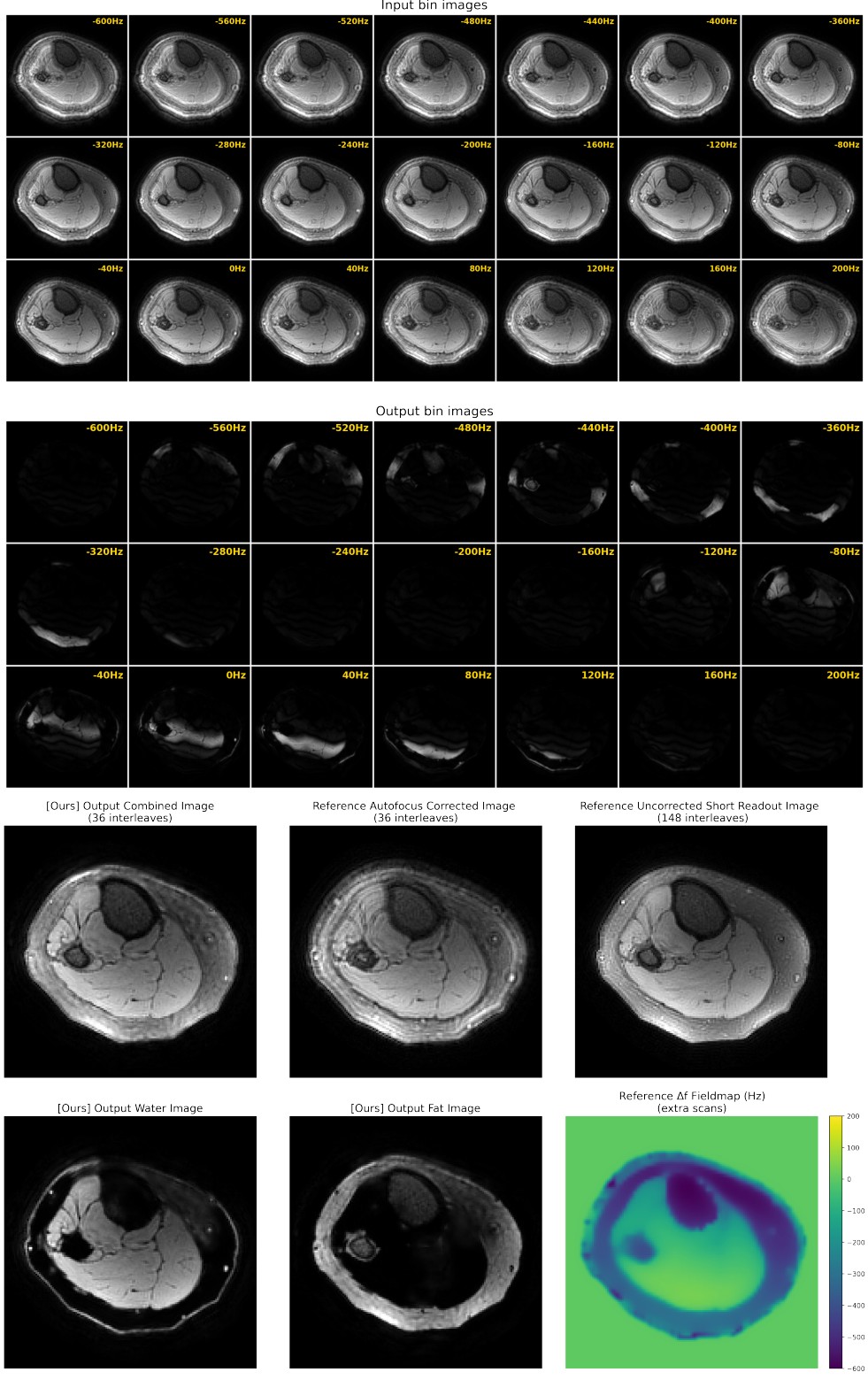

Figure 11: **Results of the proposed method on *in-vivo* PD-weighted knee scan.** Acquired using Gradient Echo (GRE) sequence (TE=4ms, TR=2000ms, FA=30°). Top: Frequency bin demodulation inputs. Middle: Frequency bin output images. Bottom: 1) output combined fat-water images, 2) corrected image using Autofocus [2], 3) reference uncorrected short-readout image (4x scan time), 4) output water image, 5) output fat image, 6) reference off-resonance frequency map.

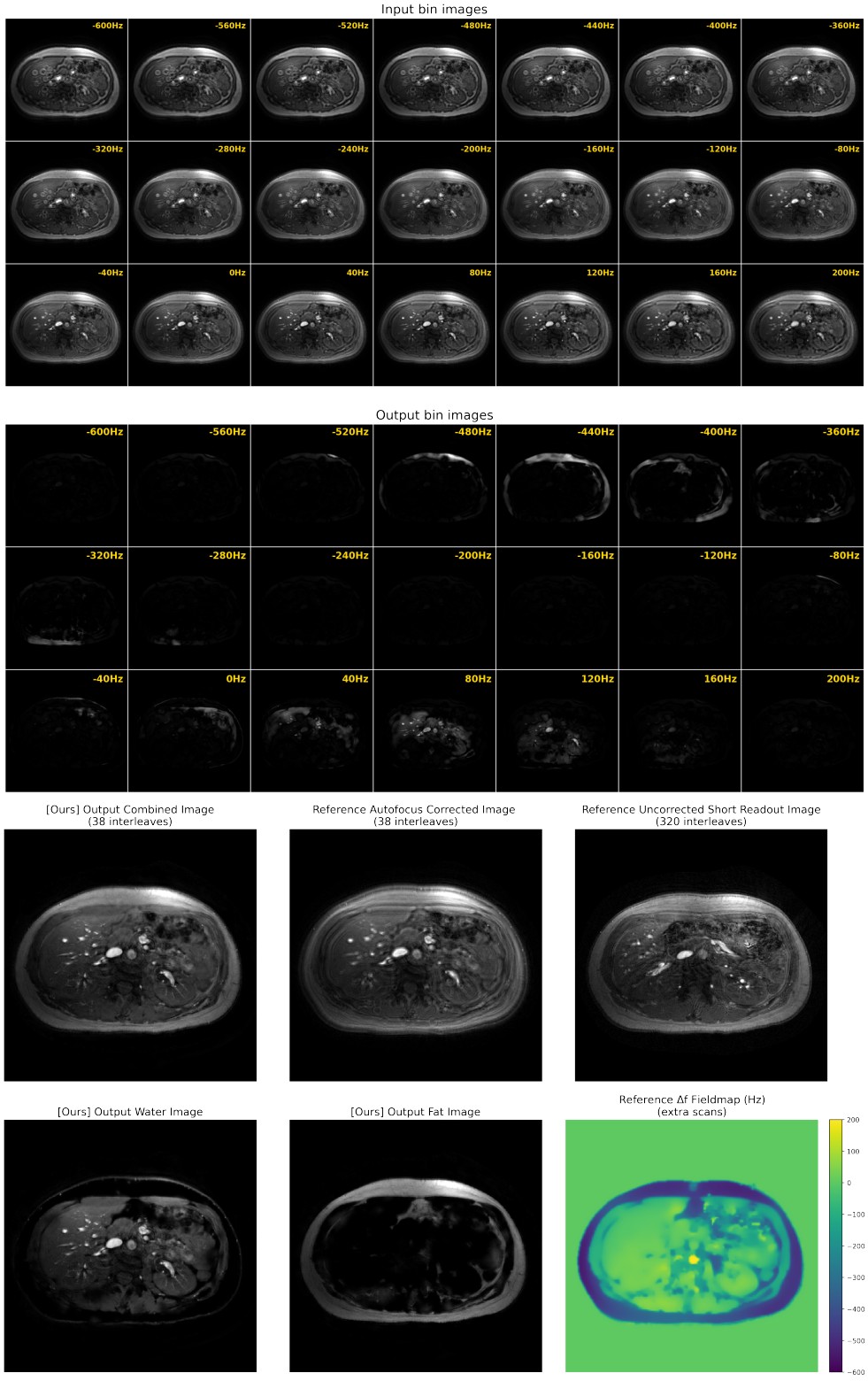

Figure 12: **Results of the proposed method on *in-vivo* T1-weighted abdominal scan.** Acquired using Gradient Echo (GRE) sequence (TE=3ms, TR=50ms, FA=60°). Top: Frequency bin demodulation inputs. Middle: Frequency bin output images. Bottom: 1) output combined fat-water images, 2) corrected image using Autofocus [2], 3) reference uncorrected short-readout image (8x scan time), 4) output water image, 5) output fat image, 6) reference off-resonance frequency map.

### B.3 Ablation Study: Omitting CNN Prior in Data Consistency

Equation 5 in the main paper describes the objective function of the Data Consistency block. Attempting to solve this function without the integration of the CNN prior results in the optimization failing to correct for off-resonance blurring. Figure 13 depicts the resultant output for an unrolled reconstruction of an in-vivo example when neglecting $D_{\phi_k}$. The outcome remains consistent regardless of whether $x_0$ is initialized as zeros or as $A^H y$. Both optimizations produce $\hat{x}$ with bins that retain off-resonance blurring artifacts. Given the inherently ill-posed nature of the problem, this outcome aligns with our expectations.

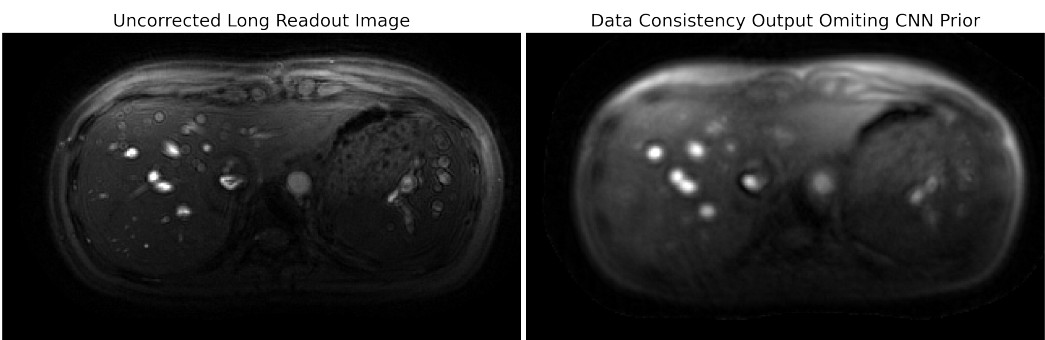

Figure 13: **Reconstruction output without incorporating the CNN prior in Data Consistency module**. Both initializations of $x_0$, whether zeros or $A^H y$, fail to correct the off-resonance blurring artifacts.