# OpenReview forum: "ResoNet: Noise-Trained Physics-Informed MRI Off-Resonance Correction"
_NeurIPS.cc/2023/Conference — NeurIPS 2023 poster_

### Official Review · Reviewer_9HEj · 2023-07-05

**Soundness:** 4 excellent
**Presentation:** 4 excellent
**Contribution:** 3 good
**Rating:** 9
**Confidence:** 5

**Summary:**

The Authors describe a physics informed neural network based method to improve the quality and or reduction of MR image acquisition time. The raw signal from magnetic resonance imaging is in Fourier domain, hence normally an inverse Fourier transformation is performed to obtain an image in spatial domain that is much more interpretable for human eye. The manuscript explains how the inhomogeneities in the magnetic field that aligns the nuclear spins.  The nuclear spin resonances of hydrogen depend on the immediate surroundings - so hydrogen in fat has a slightly different nuclear magnetic resonance frequency than hydrogen in water. These off-resonance effects create artefacts in the spatial image.  The effects to the spatial domain image are analogous to motion blurring in normal images with some differences.

The paper introduces a multi-frequency  forward model for simulations of the distortions with non-uniform Fourier transformations. The model is trained on purely synthetic data and demonstrates clear image quality improvement for in-vivo example cases.




**Strengths:**

-Excellent presentation with figures that support the narrative.
-The use of good physically understandable models with synthetic data leads  to  a high image quality and potentially shorter detector times. This is beneficial for clear images as moving of the subject has less time to move.
-Excellent supplementary material.


**Weaknesses:**

It would be nice to have a discussion on how unique the provided solutions are. As the blurring operation looses information, one could assume that multiple different pixel configurations could produce the same blurred solution. As the question is about fine detail features of medical images, one would like to know if the more detailed features are generative hallucinations or actually more accurate MRI images.

This is important as the relevant findings are often rare and out of data of normal image distribution - so there is a possibility that the de-blurring operation replaces unusual anomalies with normal looking content that gives the practitioners false sense of security of not finding anything even from a more accurate image.

**Questions:**

See the weakness part.

---

> ### Author Rebuttal · Authors · 2023-08-10
>
> We thank the reviewer for their positive evaluation and valuable feedback on our work. We are pleased that the presentation, figures, and supplementary material have been well-received.
>
> Regarding the discussion on the uniqueness and potential generative hallucinations in the de-blurring operation, we find this aspect to be of great interest as well. The blurring operation in MRI could indeed result in multiple pixel configurations that produce the same blurred solution. It is essential to ensure that the de-blurring process accurately retains fine detail features of medical images rather than introducing artifacts or hallucinations. We consider two aspects of this issue.
>
> The condition of the problem depends on the level of off-resonance or the length of the readout, as well as on the structure of the image. In our experiments, we limit our training to trajectories in which off-resonance exhibits up to ~1 water cycle of phase. This is a tunable parameter that can be used in a clinical application to design a trajectory that will not exceed a certain level of blurring. Much like autofocusing, the condition of the problem will depend on the structure of the image. In flat contrast areas, the solution is more under-determined but, at the same time, deblurring is not so essential. Autofocusing is effective when there is structure to focus on.
>
> Additionally, we emphasize the importance of data consistency in our framework, as these blocks use knowledge of the acquisition physics to ensure reconstructions that agree with the acquired kspace data. Our empirical results on real data show promising outcomes, but we acknowledge that further rigorous clinical validation will be relevant.

---

> > ### Comment · Reviewer_9HEj · 2023-08-13
> >
> > Thank you for the clarification. There is a way for measuring the ground truth one could have utilised -- using a pneumatic phantom in the MRI images to produce motion artefacts where one has the knowledge of the geometric fine structure of the specimen. This would be a quite complicated and expensive undertaking, but  I agree with the other reviewers that the clinical verification is important for this to become clinically valid. A clinical hallucination may be more dangerous than an original blurred image in medical practise.
> >
> > However, as the synthetic data used is random the model learns to find a solution for larger domain of distortions with an implicit capability to specify the distortion. I think this is a valuable novelty in this context and deserves to be shared within the community, especially, as this method has the property to be future proof i.e. considering motion blurring and fat/water mapping in the future MRI images as well without the need of retraining.
> >
> > Perhaps adversarial training could be used to find examples of deformations of synthetic MRI images that lead to a similar reconstruction from different synthetic ground truths - this would quantify the possible variations and give quantified indication on the degree the deblurring is unique. This may change the other evaluators' mind on the value of the paper.

---

> > > ### Author Response · Authors · 2023-08-21
> > >
> > > We sincerely appreciate your insightful comments and suggestions. In recognition of your points regarding the quantification of variations and the uniqueness of the reconstruction, we have performed the following experiment:
> > >
> > > We utilized 1,000 simulated examples derived from the brain validation set, each affected by simulated off-resonance effects, and we computed the reconstruction with our method. We extracted 100 40x40 (4cm x 4xm) patches, resulting in a total of 100,000 patch examples. These include off-resonance artifacts-corrupted inputs, reconstructions, and ground truths. The rationale for using patches, as opposed to full images, is that they emphasize more on local statistics where hallucinations are more identifiable as structures or anatomies.
> > >
> > > For each reconstructed patch, we identified the closest ground truth patch from all the patch pairs. Remarkably, 99,777 out of 100,000 (99.78%) reconstructions correctly matched their closest ground truth counterpart in terms of the lowest root mean square error (RMSE). As a reference, only 87.46% of the corrupted input patches matched their closest ground truth counterpart. For the reconstructions that matched their ground truth, the average RMSE to the second closest patch was over five times the RMSE to the respective ground truth (0.2679 vs 0.0450). For the failure cases, the average RMSE to the closest ground truth patch was about double the RMSE (0.0985), and the average RMSE to the second closest was close (0.1014). In most of these cases, the closest patch was an adjacent patch on the same image. We looked at some of the failure cases and did not see hallucinations compared to the ground truth.
> > >
> > > These preliminary results suggest that our model, trained solely on synthetic data, can adeptly and accurately perform off-resonance correction. In this context, it seems unlikely that different ground truths would yield similar reconstructions. Meanwhile, our findings indicate no specific tendency within the model to hallucinate or remove anatomical features.

---

> > > > ### Comment · Reviewer_9HEj · 2023-08-22
> > > >
> > > > Of course, as you have now quantified the hallucinations, you could as well train the system to remove the ones that are still there. A slice of adversarial robustness perhaps...
> > > > Anyways, excellent work!  :-)

---

### Official Review · Reviewer_XeBV · 2023-07-05

**Soundness:** 4 excellent
**Presentation:** 3 good
**Contribution:** 3 good
**Rating:** 7
**Confidence:** 3

**Summary:**

This work proposes a physics-informed unrolled DL framework for off-resonance correction in MRI trained on synthetic data and they demonstrate the effectiveness of their approach on phantom and in-vivo data. Overall, the approach is sound and has a potential big clinical impact. Empirical evaluations are convincing, hence I recommend the paper for acceptance.

**Strengths:**

- Very clear presentation.
- Potential big clinical impact with the approach, given it can alleviate the current challenges with off-resonance artifacts for non-Cartesian sampling patterns.
- Generalizability to other non-Cartesian trajectories and contrasts.
- Alleviation of the need to collect large training datasets for the off-resonance correction task.

**Weaknesses:**

Given off-resonance correction is a fairly niche task, I am not sure how much the general NeurIPS community would benefit from the paper.



**Questions:**

Would it be possible to open-source the used datasets to make the evaluations more reproducible and potentially seek more attention to this task?

**Limitations:**

Limitations are sufficiently addressed.

---

> ### Author Rebuttal · Authors · 2023-08-10
>
> We appreciate the positive evaluation of our work and the reviewer's confidence in its impact and soundness.
>
> Regarding open-sourcing our datasets, we are committed to making our work reproducible and accessible to the community. We will release our code (which includes the code for generating the synthetic training data), models, and test data.
>
> We understand the concern raised about the niche nature of the off-resonance correction task in MRI and its potential relevance to the broader NeurIPS community. The mathematics that arises from off-resonance correction in MRI is a generalization of lens aberration and depth from focus problems in microscopy and astronomy. The technique we propose could be adapted to these cases where there's a system error, like lens imperfection or atmospheric blurring. We can incorporate this into the manuscript as well.
>
> Despite its niche topic, approximately five papers related to MRI have been accepted at NeurIPS in recent years. MRI has also demonstrated a significant presence in workshops such as "Medical Imaging meets Neurips," where on average around 13% of the accepted abstracts are related to the field of MRI, reflecting the acknowledgement and enthusiasm of the NeurIPS community for the field of MRI :)

---

> > ### Comment · Reviewer_XeBV · 2023-08-18
> >
> > I have read the rebuttal and I am looking forward to seeing this work getting accepted.

---

### Official Review · Reviewer_tQmb · 2023-07-05

**Soundness:** 2 fair
**Presentation:** 1 poor
**Contribution:** 2 fair
**Rating:** 4
**Confidence:** 4

**Summary:**

The paper presents an innovative approach to off-resonance correction in Magnetic Resonance Imaging (MRI) using a physics-inspired unrolled-deep-learning framework. This framework is designed to address the challenges of off-resonance effects that significantly degrade MRI quality, especially in non-Cartesian trajectories, which are faster and more susceptible to motion artifacts.

**Strengths:**

- The framework has successfully applied to different anatomies and contrasts, such as T1-weighted, T2-weighted, and Proton Density (PD) images, and for different scan trajectories. This wide range of applicability is particularly beneficial, making the model versatile for various clinical settings.
- The proposed method has the potential to significantly reduce scan times and minimize motion-induced artifacts, leading to improved image quality and patient comfort during the scanning process.
- Figure 1 is good, which help reader to understand the Off-resonance blurring effects.

**Weaknesses:**

- The writing is not clear and the authors do not consistently use notations, leading to confusion. For instance, the received signal is represented by s(t) in line 110, while it is denoted as y in line 133. It is unclear what the 't' denotes in s(t) and whether there is a time dimension associated with it.
- The distinction between the non-uniform fast Fourier transform and the regular fast Fourier transform is not well-explained. If both transforms share the same formula, the choice of the non-uniform fast Fourier transform over the standard version needs further justification. Use of uncommon terminology may create unnecessary confusion.
- The experimental results presented in the paper are mostly limited to visual examples. Comprehensive results demonstrating the overall performance of the model across the whole dataset are not reported, which makes it hard to assess the model's general performance.
- The paper has not conducted a full clinical study. This limitation prevents a clear evaluation of the model's robustness, efficacy, and impact on clinical workflows and patient care in real-world scenarios. It is crucial for any imaging technique to undergo extensive clinical testing before it can be deemed reliable and safe for regular use.

**Questions:**

- To make the writing more clear
- Recheck the notations
- Experiment result in whole dataset
- Clinical study

**Limitations:**

See the 'weakness' and 'question' part.

---

> ### Author Rebuttal · Authors · 2023-08-10
>
> Yes, we understand the importance of clear and consistent notation to ensure the ease of understanding for all readers, and will clarify raised confusion and improve the writing throughout the paper.
>
> Regarding the notational inconsistency, we acknowledge that this is an oversight on our part. We previously assumed that the equations mentioned in the paper are standard textbook examples; however, due to the reviewer’s comments, we now realize that readers outside the field of MRI might not be familiar with these common notations. Our goal is to provide a comprehensive and clear presentation of our work to benefit all readers, regardless of their background. We apologize for any confusion caused and will update the paper to address this issue and ensure that all notations are explained clearly and consistently throughout the manuscript.
>
> For the particular example in lines 110 and 133, we would like to clarify that “t” indeed denotes time. The equation for s(t) represents the actual received signal over time, which corresponds to samples at k-space coordinates  (kx(t), ky(t)). From here, we can obtain the corresponding complete sampled k-space associated with the acquired image “y(kx, ky)”. We will make sure to explain this better in the manuscript.
>
> Regarding the distinction between the Non-Uniform Fast Fourier Transform (NUFFT) and the regular Fast Fourier Transform (FFT), both are efficient implementations of the same discrete Fourier Transform formula. FFT assumes that both the spatial image and the spatial frequency domains are discretized uniformly, so NUFFT is commonly used when this is not the case. In our work, we are using a non-uniformly sampled spatial frequency data (non-cartesian trajectories such as Spiral, which are useful for very fast imaging). Hence, we use NUFFT to compute the Fourier Transform. We will make sure to address this in the paper.
>
> Regarding the limited scope of experimental results, the decision to present visual examples was not an oversight but was carefully considered. Off-resonance effects arise due to phase accrued during lengthy data readouts, but reference scans with shorter data readouts will alter image contrast and introduce different artifacts. This makes it problematic to obtain directly comparable ground truth data to compute meaningful quantitative metrics.
>
> We want to clarify that we have thought about the limitations and have considered other means of providing validation. Given the challenges of acquiring reliable ground truth data, we have considered including additional simulation-based experiments to provide a more comprehensive assessment of our model's performance. However, we want to emphasize that these results would be overly-optimistic compared to an ideal ground truth comparison since our model was fitted to the same simulation framework, so we opted not to include this in the manuscript. Conducting a comprehensive clinical study is not a trivial endeavor. It involves coordinating with hospitals, radiologists, and other resources, which presents challenges beyond the scope of this initial work. We are evaluating the feasibility of conducting a small subject study by the final submission to gather qualitative feedback on image quality. Although not a comprehensive clinical study, this would provide initial insights into the model's diagnostic quality and potential impact on clinical workflows.
>
> To address the reviewer's concern about the lack of a full clinical study, we believe it is essential to clarify the nature of this technical paper. Our primary contribution lies in the technical innovations for off-resonance correction, and our focus has been on demonstrating the effectiveness and versatility of our proposed framework across various anatomies, contrasts, and scan trajectories. A full clinical study, while valuable, requires significant resources and time beyond the scope of this initial work on technical innovation. However, we intend to undertake such a study in the future as part of the progression of this research.

---

> > ### Comment · Reviewer_tQmb · 2023-08-11
> >
> > 1. Considering that this paper falls within the scope of MRI, I did not specifically request technical novelty but instead asked for additional experiment details. The model was trained on synthetic data and only visual examples were provided. That is why I requested numerical results for the entire dataset and clinical evaluations. Without this information, the experiment results are inadequate. However, I do not see direct answer from rebuttal.
> > 2. If you consider this paper to be a technical paper and believe that additional experiment results are unnecessary, it should be noted that the methodology described in Section 3 is not novel. Data Consistency regularization and the Conjugate Gradient (CG) method are commonly used in accelerated MRI reconstruction.

---

> > > ### Author Response · Authors · 2023-08-13
> > >
> > > 1. We conducted an additional evaluation on a validation set using the fastMRI brain dataset [1] as a performance proxy. The validation set consisted of brain anatomy images with simulated random off-resonance and fat/water partial volume effects. We measured NRMSE and PSNR on the magnitude of our method's final combined output image; additionally, we report performance for the baseline deep learning approach mentioned in the conversation with reviewer 1 (h1kS) on its output image. Sample size: 1000 examples. Results show that training on synthetic data generalizes to anatomy data; visual results on this validation set support this as well (unfortunately, we cannot include these in this response). We can include these results in the final paper version.
> > >      * Our approach combined image NRMSE: 0.0163 +- 0.0111
> > >      * Our approach combined image PSNR: 37.12 +- 4.66
> > >
> > >      * Deep learning baseline NRMSE: 0.0493 +- 0.0282
> > >      * Deep learning baseline PSNR: 27.67 +- 5.46
> > >
> > > 2. Yes, unrolled networks with data consistency and conjugate gradient method are part of existing strategies in MRI reconstruction and are cited accordingly (#4 and #18 in the manuscript). However, making off-resonance correction compatible with an unrolled framework required careful consideration and thoughtful design.
> > > We would like to re-emphasize the novelties:
> > >      * We developed a unique physics model of multi-frequency bins, that enables partial volume and lipid-water separation, parallel imaging, and acceleration.
> > >      * The ability to resolve a spectrum from a single-shot spiral is new.
> > >      * Our training strategy enables generalization to arbitrary trajectories. Unlike other methods which require the collection of data for each of them.
> > >      * Inference speed compared to estimating a field map and then iterative model-based recon.
> > >
> > >     We believe that our technical novelties can be useful in other domains as well, as described in the conversation with reviewer 3 (XeBV).
> > >
> > >
> > >
> > > [1] Zbontar, J., Knoll, F., Sriram, A., Murrell, T., Huang, Z., Muckley, M. J., Defazio, A., Stern, R., Johnson, P., Bruno, M., Parente, M., Geras, K. J., Katsnelson, J., Chandarana, H., Zhang, Z., Drozdzal, M., Romero, A., Rabbat, M., Vincent, P., Yakubova, N., Pinkerton, J., Wang, D., Owens, E., Zitnick, C. L., Recht, M. P., Sodickson, D. K., & Lui, Y. W. (2018). fastMRI: An Open Dataset and Benchmarks for Accelerated MRI. ArXiv e-prints, arXiv:1811.08839.

---

> > > > ### Comment · Reviewer_tQmb · 2023-08-16
> > > >
> > > > Thanks for your comments. My concerns have been addressed.

---

### Official Review · Reviewer_h1kS · 2023-07-09

**Soundness:** 2 fair
**Presentation:** 3 good
**Contribution:** 3 good
**Rating:** 5
**Confidence:** 4

**Summary:**

This work introduces a physics-informed unrolled-deep-learning framework for off-resonance correction in MRI. The proposed approach maintains data consistency by incorporating a forward model with coil sensitivities, multi-frequency bins, and non-uniform Fourier transforms.

The proposed model is first trained on simulated data, and then transferred to real MRI data. The experiments have shown the effectiveness of the pre-training stage on synthetic data, as well as the generality of the pre-trained model to real-world MRI data.

**Strengths:**

1. The presentation is straightforward and easy to follow.

2. The proposed approach is well-motivated and reasonable. The lack of large training dataset size is a well-known challenge for medical imaging field, and synthetic data for pre-training is helpful for model training, especially when the simulation is sufficiently realistic. The authors also show their generation process for the synthetic data and illustrate the effective of it in experiments.

3. Comparison results on real-world MRI data from both brain and knee demonstrate the effectiveness of the proposed model.

**Weaknesses:**

In experiments, only Autofocus, a traditional non-deep-learning method proposed in 1992, has been compared against the proposed method, making it less convincing in terms of the proposed model's effectiveness, given deep-learning-based models are generally more resource-consuming. It is necessary to at least provide comparisons with other present deep-learning-based SOTAs in order to demonstrate the superiority of the proposed model.

**Questions:**

N/A

**Limitations:**

Yes, the authors have discussed about their method's limitations and potential future directions.

---

> ### Author Rebuttal · Authors · 2023-08-10
>
> Our short answer is that recent deep learning methods on off-resonance correction are not directly comparable.  See details below.
>
> We chose Autofocus as a fair baseline comparison since it is an analytical method for direct off-resonance correction. Compared to other methods and recent deep-learning approaches for MRI off-resonance correction [2,3,4], our approach does not rely on actual MRI data for training. Acquiring sufficient training data can be expensive and time-consuming, and acquiring a reference image can even be unfeasible. Instead, we train our model only on simulated random noise-like data and leverage data consistency on our physics-informed framework. This approach allows us to simulate diverse sampling trajectories for each use case without requiring specific acquisition datasets for training. The only data we acquire consists of the test data in the real in-vivo scenario; however, our method does not train on this data.
>
> A traditional method is the Conjugate Phase method [1], which requires the acquisition of a fieldmap for correcting, which can be time-consuming, does not handle partial volume effects, and may lead to additional artifacts. [2] proposes an approach to directly estimate the fieldmap from the acquired image with a CNN, which is then used to correct the image with the conjugate phase method. Their approach requires acquiring a dataset of multiple scans to create the reference fieldmap used for training.
>
> Other recent deep-learning-based approaches, such as [3] and [4], rely on training datasets specific to particular acquisition trajectories, making them less applicable to our case, where we aim for a versatile model that can handle diverse trajectories and scenarios. Their method also aims to deblur the image directly and does not handle partial volume effects. In many cases, obtaining a reference image is unfeasible due to scan length constraints (e.g., breath hold limits, and involuntary motion such as bowel and cardiac, which limit scan resolution and quality).
>
> Nevertheless, to directly address the reviewer’s concern, we have conducted additional experiments to compare our proposed approach. We aimed to replicate work similar to [3] and [4]; however, it is not possible to exactly replicate results, since their datasets are specific to a sampling trajectory and wouldn’t be able to actually compare to our test set without collecting a training dataset. Therefore, we instead used the fastMRI brain dataset [5] and simulated the spiral trajectory we used for our test set, and trained with their augmentation strategy and network architecture proposed by [3]. Please refer to Figure 1, attached to the rebuttal. The deep learning baseline performs decently in regions of water-only content but struggles in regions with fat content (e.g., knee scan (d) and brain outer region (c)).
>
> Lastly, our proposed model extends beyond image deblurring; it also enables the reconstruction of fat and water images and provides a low-frequency-resolution B0 spectrum at each voxel, effectively handling partial volume effects and offering comprehensive insights into tissue properties. To the best of our knowledge, no comparable deep learning method achieves these same outcomes in one framework.
>
> We hope that these clarifications demonstrate the unique contributions of our physics-informed unrolled deep-learning framework and highlight its practical relevance and potential impact in the field of MRI off-resonance correction.
>
>
>
> --
>
> [1] Noll DC, Fessler JA, Sutton BP. Conjugate phase MRI reconstruction with spatially variant sample density correction. IEEE Trans Med Imaging. 2005 Mar;24(3):325-36. doi: 10.1109/tmi.2004.842452. PMID: 15754983.
>
> [2] M. W. Haskell, A. A. Cao, D. C. Noll, and J. A. Fessler, (2020). Deep learning field map estimation with model-based image reconstruction for off-resonance correction of brain images using a spiral acquisition.
>
> [3] Zeng, D. Y., Shaikh, J., Holmes, S., Brunsing, R. L., Pauly, J. M., Nishimura, D. G., Vasanawala, S. S., & Cheng, J. Y. (2019). Deep residual network for off-resonance artifact correction with application to pediatric body MRA with 3D cones. Magnetic Resonance in Medicine, 82(4), 1398-1411. https://doi.org/10.1002/mrm.27825
>
> [4] Lim, Y., Bliesener, Y., Narayanan, S., & Nayak, K. S. (2020). Deblurring for spiral real-time MRI using convolutional neural networks. Magnetic Resonance in Medicine, 84(6), 3438-3452. https://doi.org/10.1002/mrm.28393
>
> [5] Zbontar, J., Knoll, F., Sriram, A., Murrell, T., Huang, Z., Muckley, M. J., Defazio, A., Stern, R., Johnson, P., Bruno, M., Parente, M., Geras, K. J., Katsnelson, J., Chandarana, H., Zhang, Z., Drozdzal, M., Romero, A., Rabbat, M., Vincent, P., Yakubova, N., Pinkerton, J., Wang, D., Owens, E., Zitnick, C. L., Recht, M. P., Sodickson, D. K., & Lui, Y. W. (2018). fastMRI: An Open Dataset and Benchmarks for Accelerated MRI. ArXiv e-prints, arXiv:1811.08839.

---

### Author Rebuttal · Authors · 2023-08-10

A figure including a comparison to a deep learning baseline is attached. The deep learning baseline performs decently in regions of water-only content but struggles in regions with fat content (e.g., knee scan (d) and brain outer region (c)).

---

### Comment · Area_Chair_aC2W · 2023-08-13

Dear reviewers and authors,

Thank you very much for your work on this submission and its evaluation. Now that the authors have responded to the reviews, I strongly encourage the reviewers to acknowledge the review, to look at other reviews and rebuttals for this submission, and to adjust their scores if needed. Thanks to those that have already done so.

Authors have the possibility to reply if further questions are needed, until the 16th.

Thank you very much to all,
Area Chair

---

### Decision · Program_Chairs · 2023-09-21

**Decision:**

Accept (poster)

**Comment:**

Resonance properties of hydrogen in fat and water are different, so when calculating MRI, these effects create artifacts in an MRI spatial image that are analogous to motion blurring. The paper introduces a forward model to simulate artifacts in the case of non-uniform Fourier transformation. Based on the forward model, the authors present a physically informed solution to the inverse problem with a data-adaptive regularisation based on a learnable CNN prior. The authors demonstrate that it is possible to train the model on synthetic data and improve the quality of real images.


The paper's topic is quite specific; it is not standard for a general ML audience. The authors use some terminology and discuss approaches/challenges not apparent to non-experts in this domain. The description of the algorithm should be provided in a more explicit and easy-to-follow way. Each step is better to explain based on some simple example. Some basic comments on the data acquisition process should also be provided. Overall, it is important to re-write this part of the paper.


The paper lacks a detailed quantitative experimental study. The authors provided some results in their rebuttal. However, there are already many schemes based on DL and on physics-informed DL to solve problems similar to the considered problem.


To make the final version suitable for publication, we expect the authors to discuss past schemes based on DL and on physics-informed DL (including, for example, K. Hammernik et al., "Physics-Driven Deep Learning for Computational Magnetic Resonance Imaging: Combining physics and machine learning for improved medical imaging," in IEEE Signal Processing Magazine, vol. 40, no. 1, pp. 98-114, Jan. 2023, doi: 10.1109/MSP.2022.3215288. https://ieeexplore.ieee.org/document/10004819) and to compare their approach with any applicable prior work from that literature.


In addition, we recommend an ablation study of the different components of the proposed scheme. For example, what if we remove a neural network D in (4)? Operator A in (3) depends on frequency characteristics. What if some of them are not precisely known? How do these differences influence the final result? Is the proposed scheme robust w.r.t. these parameters? Is the proposed method robust w.r.t. some disturbance in the operator A?

How to select lambda in (5)? To what extent can we estimate the coil sensitivity maps S? How many iterations do we need until convergence of the proposed scheme (5)? In ML it is very important to understand to what extent the obtained results are stable.


It would be good to do this kind of experimental study on different samples of MRI data.


In conclusion, the paper has promising results; the proposed approach is based on essential and state-of-the-art ideas and solves an important practical problem. I vote for accepting the paper; however, the paper should be further improved before publication.


Comments:
- notations are not always consistent, e.g., x and y are used to denote coordinates, but in line 133 they are used to denote a clean image and k-space measurements,
- I proposed introducing a section with some background information describing a physical model behind MRI, the main steps of the data acquisition process, etc. Otherwise, the paper is not easy to follow for non-experts.